# Optimal Design of an Inductive MHD Electric Generator

Sara Carcangiu [ID], Alessandra Fanni and Augusto Montisci *[ID]

Electrical and Electronic Engineering Department, University of Cagliari, Via Marengo, 2, 09123 Cagliari, Italy
* Correspondence: augusto.montisci@unica.it

**Abstract:** In this paper, the problem of optimizing the design of an inductive Magneto-Hydro-Dynamic (MHD) electric generator is formalized as a multi-objective optimization problem where the conflicting objectives consist of maximizing the output power while minimizing the hydraulic losses and the mass of the apparatus. In the proposal, the working fluid is ionized with periodical pulsed discharges and the resulting neutral plasma is unbalanced by means of an intense DC electrical field. The gas is thus split into two charged streams, which induce an electromotive force into a magnetically coupled coil. The resulting generator layout does not require the use of superconducting coils and allows you to manage the issues related to the conductivity of the gas and the corrosion of the electrodes, which are typical limits of the MHD generators. A tailored multi-objective optimization algorithm, based on the Tabu Search meta-heuristics, has been implemented, which returns a set of Pareto optimal solutions from which it is possible to choose the optimal solution according to further applicative or performance constraints.

**Keywords:** static energy conversion; non equilibrium plasma; multi-objective optimization; Tabu Search; Pareto Front

## 1. Introduction

Magneto-Hydro-Dynamic (MHD) electric generators have seen a great effort in their development, thanks to the advantage of carrying out thermoelectric conversion without any solid part in motion [1,2]. This feature implies several advantages, but at the same time some shortcomings remain, despite the great dedicated research efforts.

The operating principle of the MHD electric generator is quite simple: a conducting fluid (plasma) flows through a magnetostatic field, becoming the seat of an induced electromotive force, which will be the greater the speed of the fluid and the greater the intensity of the external magnetic field. If two electrodes are placed in direct contact with the plasma and connected to an electrical load, the plasma is crossed by a current which will be all the more intense the lower the resistivity of the plasma. It follows that the efficiency of the MHD generator basically depends on three parameters: conductivity of the plasma; intensity of the applied magnetic field; speed of the plasma.

When the operating fluid is not a good electrical conductor, as in the case of combustion products, its conductivity must be increased. This is achieved both by increasing the temperature of the fluid [3] and by adding the fluid with conductive materials [1], such as Potassium, Sodium, Caesium, Aluminium. Thanks to seeding it is possible to obtain a sufficient value of the conductivity of a fluid at temperatures around 1800 °C. This means that the fluid leaves the generator with a high enthalpy content, so the generator must necessarily be thought of as the topping stage of a combined cycle. At the same time, the problem of seeding recovery arises, as the seeding materials cannot be dispersed into the environment due to their toxicity and/or economic value.

Another limitation is due to the need to have very intense magnetic fields (5 T) which require a superconducting coil system. For this reason, the system to produce the magnetic field, also including the cryogenic system, constitutes the most complex and expensive part of the entire apparatus.

Finally, the electrodes, with which the current is drawn, are subject to rapid deterioration due to both high temperatures and current values, and this was probably the factor that most affected the decline in interest in MHD generators that has been recorded since the end of the last century.

To overcome these drawbacks, several studies have been made, with the aim of exploiting magnetic induction to transfer energy from plasma to armature, without the need for electrodes [4–8].

In [4] one of the first studies on inductive MHD generators is reported. The aim of the study was to determine the power generation characteristics of a generator in which the plasma was generated in a conventional diaphragm-type shock tube by inductively coupling the load to the magnetic induction field generated by the currents flowing in the plasma. Preliminary results showed that the studied system was not yet a practical current generator as a very low output power was obtained compared to that which had been obtained up to that moment by electrode type MHD generators.

An improvement of the induction MHD generator is obtained in [5] by reducing the eddy current, and therefore the losses. In the patented generator, the walls of the flow channels have been divided into individual electrically isolated sections. This causes a reduction in the magnetic flux induced voltage per wall section and a corresponding reduction in the eddy currents.

In [6] the induction is obtained by means of a traveling magnetic field generated by a stator winding. The principle of functioning is the same of the asynchronous generator, and it provides power to the network if the velocity of the plasma is higher than the velocity of the stator field. In [7] an alternate generator is proposed, where the induction is caused by the variation of the velocity over time. This layout allows to use MHD generation to harvest energy from environment, such as sea waves, but it is not directly applicable in a combustion cycle.

More recently, an alternate MHD induction generator has been proposed [8], combined with a thermoacoustic machine, which converts thermal energy into mechanical vibration. The plasma consists of a liquid metal and the magnetic field is generated by a permanent magnet. As the vibration is transmitted to the liquid metal, this becomes seat of an induced alternate current, which operates as the primary winding of a transformer, transmitting the power to a secondary winding magnetically linked. In [9], a similar system is proposed, but in this case the liquid metal is replaced by an ionized gas, whose charge carriers are kept separate by means of an electric field.

In [10] an inductive MHD generator layout is proposed, where the induction is produced by using a non-equilibrium plasma, whose charge concentration changes over time giving rise to a variable electrical current. The variable current induces an electromotive force on a toroidal coil wrapped around the duct, which supplies the electrical load.

In [11] a one-dimensional model of the MHD electric generator proposed in [10] has been implemented in Simulink® environment and some simulation results, which are in good agreement with the expected ones, are presented. Moreover, in [11] some preliminary indications for the device dimensioning are given, concerning the density of charges, the velocity of the gas, the size of the core, and the cross section of the duct. However, a more thorough investigation of the design of the device is needed, which is the purpose of the present paper.

The proposed generator layout overcomes the MHD generator restrictions such as gas conductivity and electrode corrosion without the need of superconducting coils, whose scientific benefits are of great importance. However, without an appropriate dimensioning of the generator, the performance of it would be reduced for an actual use. This paper gives a contribution in this direction evaluating a set of Pareto optimal solutions from which the designer can choose the best one, depending on other constraints.

In particular, the optimal design of the MHD generator requires to find the design parameters (total charge, velocity of flow, toroidal core dimensions, number of turns in the coils) that realize a compromise among conflicting objectives (mass of the device, viscous

losses, output power). Therefore, a Multi-Objective Optimization (MOO) [12] is mandatory. As usual in MOO techniques, the aim of the optimization is not to find the best solution, but rather to generate the subdomain of the non-dominated solutions, lying on the Pareto Front [13]. In most cases, the analytical expression of the Pareto Front is not attainable, therefore the MOO will provide a sampling of such subspace.

The design of the inductive MHD electric generator has been formalized as a multi-objective optimization problem and solved by using a fully-vectorial multi-objective Tabu Search (MO-TS) [14], able to sample the Pareto Front on an almost regular grid.

The non dominated solutions in the obtained Pareto Front have been analyzed and further fitness function have been introduced to choose that one suitable for use in specific applications such as spatial and automotive, or industrial applications.

The paper is organized as follows. In Section 2, the principle of functioning of the inductive generator is described. In Section 3, the optimization procedure is formalized. Section 4 reports the implementation details and some Pareto optimal solutions in terms of design parameters and corresponding objective functions, and the pros and cons of the different solutions are discussed. Finally, some conclusions, comments and future works are given in Section 5.

## 2. Principle of Functioning of the Inductive MHD Generator

Figure 1 describes the principle of functioning of the proposed inductive MHD electric generator. A gas coming from a combustion chamber is ionized by means of periodical pulsed electric discharges, and, successively, a DC electrical field, applied to the gas using the plates of a capacitor, separates the positive charge carriers from the negative ones. The main stream is split into two electrically unbalanced streams, giving rise to a time-variant electric current loop (in yellow in Figure 1). This current induces an electromotive force into two toroidal coils (see the same Figure 1), which supply the electric load. The time-variant electric current depends on several design parameters, such as the rate and the intensity of the discharges, the external electric field, the velocity of the gas, the mobility of the charge carriers.

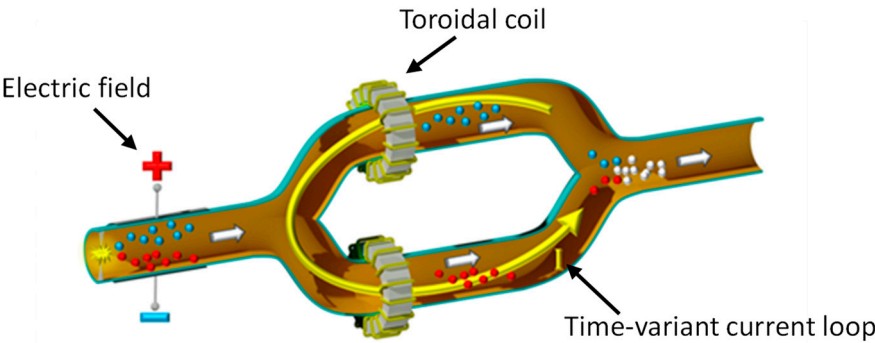

**Figure 1.** 3D layout of the Inductive MHD electric generator. In yellow the time-variant current loop.

The generator works as a current transformer, where the primary circuit is constituted by the duct where the plasma flows, while the secondary circuits are the toroidal coils connected to the electric load.

This layout allows us to avoid the above-mentioned limits of classical MHD generators: the ionization of the gas is no longer due to the temperature or to the seeding, the electrodes are no longer required to retrieve the current from the gas, and, finally, the excitation field is electrostatic, rather than magnetic, therefore superconducting coils are no longer needed. On the other hand, new technological issues arise. In particular, as the conversion process is based on induction, the flow of charges through the toroidal ring must be time-variant. This is performed by operating the device in pulsed way and generating clouds of charges as compact as possible. However, the mutual repulsive force among charges of the cloud has negative effects as it tends to spread the charges along the axis of the duct, and it leads

to prevent any conversion when the charges are uniformly spread. To mitigate this effect, it is important to make the velocity of the gas as fast as possible, and to minimize the electrical mobility of the charge carriers, for example by increasing the level of pressure. However, these measures have also negative effects in terms of hydraulic losses.

The magnetic flux, which circulates in the toroidal core, has a non-zero average, which reduces the efficiency with which the magnetic material of the core is used. In fact, since the continuous component of the flux does not contribute to induction, the continuous component is a drawback, as the only effect is that one of saturating needlessly the magnetic material. To avoid this effect, it is possible to substitute the DC exciting field with an alternate one, whose frequency is one-half of the rate of discharges. In this way, the primary current (yellow line in Figure 1) becomes alternate, and then the average flux is zero. This solution does not come without consequences. As said above, one of the key factors of the process is the capability to maintain the charge clouds as compact as possible. In case the charge carriers that pass in one duct have all the same sign, the mutual repulsion between consecutive clouds mitigates the tendency of the single cloud to spread along the duct. If, instead, positive and negative clouds alternate in the same section, the spreading of the clouds along the duct is accentuated, and then the velocity of gas should be increased, or the rate of electrical discharges reduced, in such a way that two consecutive clouds do not interact.

Another important issue is related to the mass of the device. The induction effect depends on the reluctance of the magnetic circuits which should be minimized, hence on the permeability of the magnetic cores, that must be as high as possible. For a given material, this achievement can be obtained just by increasing the cross section of the magnetic cores. Nevertheless, depending on the specific application, very strict constraints could limit the feasible value of the device mass. This holds true, for example, in automotive and space applications, where the inertia of the vehicles has a fundamental importance. Therefore, the designer must seek the best compromise among conflicting objectives to find an optimal solution.

The operation of the device is also affected by the armature reaction. As can be observed in Figure 2, the ring of varying magnetic flux $\Phi$ in the toroidal coil generates an electric field $E$, whose force lines are identical to the force lines of the magnetic field generated by a ring of current in the same position of the magnetic flux $\Phi$ [9,11].

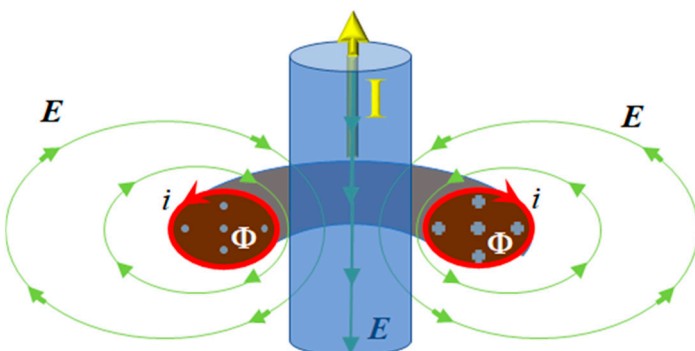

**Figure 2.** Magnetic circuit (grey: magnetic core; red: current *i* in the coil; $\Phi$: grey points and crosses); duct (light blue); electric field *E* (green lines); primary ionic current I in the duct (yellow).

The existence of the electric field $E$ can be demonstrated by following the same procedure by which the Biot-Savart law is obtained. To this end, starting from the definition of the potential vector **A**:

$$\mathbf{B} = \nabla \times \mathbf{A} \tag{1}$$

a new potential vector **C** is defined as follows:

$$\mathbf{A} = \nabla \times \mathbf{C} \tag{2}$$

from which:

$$\mathbf{B} = \nabla \times (\nabla \times \mathbf{C}) = \nabla(\nabla \cdot \mathbf{C}) - \nabla^2 \mathbf{C} \tag{3}$$

By assuming a solenoidal field $\mathbf{C}$ ($\nabla \cdot \mathbf{C} = 0$) it results:

$$\mathbf{B} = -\nabla^2 \mathbf{C} \tag{4}$$

which is the Poisson's equation, whose general solution writes:

$$\mathbf{C}(r) = \iiint_V \frac{\mathbf{B}}{r} dV \tag{5}$$

where $r$ is the radius vector of the point where $\mathbf{C}$ is evaluated, and $V$ is the volume of the torus, as the induction $\mathbf{B}$ is assumed not null only within the iron.

Moreover, its value is assumed constant in the cross section of the torus. Therefore, in a generic point near the core, it writes:

$$\mathbf{C}(r) = \oint_L \frac{\Phi}{r} d\boldsymbol{l} \tag{6}$$

where $\Phi$ is the magnetic flux in the core (Figure 2), $L = 2\pi R$ is the midline of the torus, and $R$ is its radius. Equation (6) can be interpreted in differential terms, as follows

$$d\mathbf{C}(\boldsymbol{r}) = \frac{\Phi}{r} d\boldsymbol{l} \tag{7}$$

By calculating the curl of Equation (7), it gives the differential term of the vector potential $\mathbf{A}$

$$\nabla \times (d\mathbf{C}) = d\mathbf{A} = \Phi \cdot \frac{d\boldsymbol{l} \times \boldsymbol{r}}{r^3} \tag{8}$$

The integral of Equation (8), calculated along the midline of the torus, gives the potential vector in the generic point of the space around the torus. The derivative of the potential vector with respect to time gives the electrical field:

$$E_\Phi(r) = -\frac{\partial A(r)}{\partial t} = -\frac{\partial}{\partial t} \oint_L \Phi \cdot \frac{dl \times r}{r^3} \tag{9}$$

In particular, the electric field in the point $x$ of the axis of the torus writes:

$$E_\Phi(x) = -\frac{2\pi \cdot R^2}{\left(\sqrt{R^2 + x^2}\right)^3} \cdot \frac{d\Phi}{dt} \tag{10}$$

The armature reaction of the device originates from the field $E_\Phi$. When the coil is open, i.e., no electrical load is connected, the flux $\Phi$ is generated only by the ionic current $I$, and then the field $E_\Phi$ is in quadrature with respect to the ionic current, which implies that the average transfer of power is null. On the contrary, when the coil is closed on an electrical load, the flux $\Phi$ is due to the superposition of the two contributions of the ionic motion $I$ and of the current $i$ circulating in the coil. Hence, the flux $\Phi$ is no longer due to the ionic current only, and it is no longer in phase with the ionic current. The phase shift of the magnetic flux makes it possible to transfer part of the enthalpy of the gas to the electrical load. In this last case, the system performs as a transformer, while in the previous case the toroidal coil performs as an inductor.

Furthermore, this reaction effect is multiplied by the number of turns, but a minimal number of turns is necessary to obtain a suitable voltage, then the number of turns should represent a tradeoff solution between these two conflicting requirements. A reduction of the armature reaction can be also obtained by increasing the impedance of the electrical load, so that the current responsible of the reaction is reduced. To this end, and without

affecting the level of power, one transformer can be interposed between the generator and the electrical load.

## 3. Optimal Design of the Generator

The optimal design of the generator would require that all the design parameters can be modified. On the other hand, as explained in the previous section, an equivalent effect can be obtained by changing different parameters. This fact allows us to perform the optimization of the generator in a space whose dimensions are less than the design parameters.

For the sake of simplicity, only one section of the duct, in correspondence of one of the two coils, is analyzed, by considering separately the case of positive and negative charge carriers. The clouds of charges follow each other at the rate of electrical discharges, and they change both shape and position depending on the applied forces. Such forces are the dragging force of the neutrals, the mutual repulsion between charges of the same sign and the armature reaction due to the alternate current circulating in the coils. The chosen variables under control in the optimization process are the total charge, the velocity of the gas, the size of the core and the number of turns of the coil. Other design parameters have been fixed empirically.

In Figure 3, the schematic view of the MHD section to be optimized is shown.

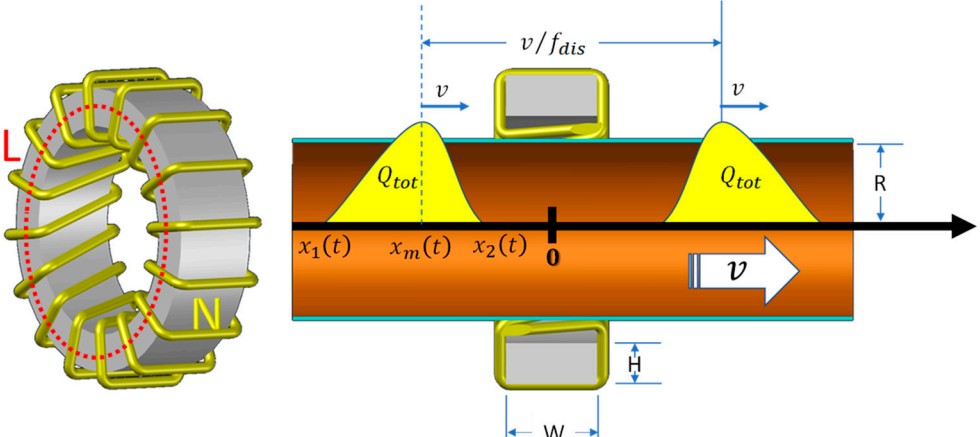

**Figure 3.** Schematic view of the MHD section to be optimized ($f_{dis}$ is the rate of the discharges). In the left side, the 3D toroidal coil is depicted.

### 3.1. Multi-Objective Optimization Algorithm

As described in the previous section, the objectives of the design problem are conflicting. Hence, there is not a single solution that meets all the objectives, but several possible trade-off solutions, known as Pareto-optimal set or Pareto Front. The role of a multi-objective optimization algorithm is to find the best possible representation of the Pareto Front. In literature, several multi-objective optimization algorithms have been proposed for the optimal design of complex devices, belonging to the heuristic search space algorithms such as Genetic Algorithms(GA) [15], Tabu Search (TS) [14], fuzzy logic [16,17], neural networks (NN) [18–21] and, more recently, hybrid algorithms [16,22]. In [18], a multi-objective TS has been implemented to optimize the design of an electromagnetic device, where a NN has been used to approximate the objective functions that depend on the design variables, whereas in [19], a neural network is trained to learn the relationship between the design parameters of a linear compliant mechanism of nanoindentation tester and the output performance based on a training set built using Central Composite Design and Finite Element Method (FEM). A GA is then used to obtain the Pareto optimal set. Similarly, in [20,21], the NN is used to predict the performance of a pump with different design parameters, and GA is used to find the optimal model. In [16] a multi-objective

optimization design of a compliant gripper mechanism as a robot arm has been presented combining fuzzy logic and adaptive neuro-fuzzy inference system (ANFIS). In [17], a training algorithm based on the non-symmetric fuzzy means (NSFM) approach, is hybridized with the TS to produce power curve models of wind turbines. In [22], a multi-objective hybrid algorithm, based on GA and TS has been presented to minimize make-span in flow shop scheduling problem.

Due to the complexity to evaluate the objective functions in the MHD generator, and due to the high number of design variables that characterizes our optimization problem, an exact solution of the problem is not affordable within a reasonable computation time.

In the present work, a multi-objective search algorithm based on the Tabu Search metaheuristics [23] has been implemented (MO-TS), in which a one-dimensional model developed in Simulink$^{®}$ environment has been used for the evaluation of the objective functions [11], avoiding the use of approximators such as NN models or fuzzy approaches. These lasts are mandatory in the large majority of usual engineering problems where, e.g., the Finite Element Method is used and models with a large number of design parameters are required.

Tabu Search (TS) basically consists of moving into the design parameters domain from a current solution $x_c \in \mathbb{R}^n$ to another one belonging to its neighborhood that improves a fitness criterion. Intensification and diversification strategies are implemented to direct the search of the optimal design solution in the most promising regions or to escape from less promising regions exploiting the history of the search following a reactive schema [24,25]. To this purpose, a Tabu List (TL) is introduced containing the solutions already visited, marked as taboo for a number $TT$ (Tabu Tenure) of iterations which depends on the length of the TL. TS was initially designed for combinatorial problems, where the design variables are of discrete nature [23]. To apply the basic concept of TS to continuous variables, the variables domain can be sampled in a uniform *n*-D grid of points, which becomes the new search domain. A local search can then follow to explore more finely the neighborhood of the current solution, as proposed in [25].

A multi-objective optimization problem can be formalized as:

$$\min_{x \in S} F(x); S = \{x \in \mathbb{R}^n | G(x) < 0; H(x) = 0\} \tag{11}$$

where $x \in S$ is the vector of the $n$ design variables and $F(x) \in \mathbb{R}^m$ is the vector of $m$ objective functions. $G(x)$ and $H(x)$ are the problem constraints, whereas $S$ is the set of feasible solutions.

### 3.2. Evaluation of the Objective Functions

At the basis of this study is the assumption of being able to study electromagnetic phenomena independently of fluid dynamics. This is done by assuming, as input value, the velocity of the neutral particles of the plasma, which derives from the dynamic balance of the forces involved. In this way, the two studies can be conducted independently each other. Note that, the fluid dynamics inverse problem, which is not treated here, consists of determining the pressure value that produces the velocity assumed by hypothesis.

Starting from this assumption, the objective functions are evaluated by integrating the following nonlinear first-order differential equation, which represents the Ampère's law applied to the torus midline [11]:

$$\oint_L H \cdot dL = I + N \cdot i + \varepsilon_{gas} \frac{d}{dt} \iint_{S_{duct}} (E_\rho + E_\Phi) dS_{duct} \tag{12}$$

where the magnetic field $H$ is integrated along the torus midline $L$, $I$ is the ionic current crossing the torus, $N$ is the number of turns of the coil, $N \cdot i$ is the conduction current contribution of the coil. The last term $\varepsilon_{gas} \frac{d}{dt} \iint_{S_{duct}} (E_\rho + E_\Phi) dS_{duct}$ is the displacement current in the origin due to the sum of the Coulomb's field $E_\rho$ and the field $E_\Phi$ induced by

the time-varying flux $\Phi$, $\varepsilon_{gas}$ is the permittivity of the gas, and $S_{duct} = \pi R^2$ is the section of the duct, where $R$ is the radius of the duct cross-section. The Equation (12) can be re-written in terms of the unknowns $\Phi$ and the charge distribution $\rho(x, t)$:

$$\oint_L H \cdot dL = \frac{L}{\mu_{core} A} \Phi \tag{13}$$

$$I(0, t) = \rho(0, t) \cdot \left[ v + m_E \left( E_\rho(0, t) + E_\Phi(0, t) \right) \right] \tag{14}$$

$$i = \frac{N}{R_{el}} \frac{d\Phi}{dt} \tag{15}$$

$$E_\rho(0, t) = -\frac{1}{2\varepsilon_{gas}} \int_0^R \int_{x_1(t)}^{x_2(t)} \frac{\gamma \cdot \xi}{[\xi^2 + \gamma^2]^{\frac{3}{2}}} \rho(\xi, t) \, d\gamma \, d\xi \tag{16}$$

$$E_\Phi(0, t) = -\frac{2\pi}{R} \frac{d\Phi}{dt} \tag{17}$$

where $\rho(x, t)$ is the linear charge density at the reference point $x$ at time $t$ ($\rho(0, t)$ is the charge density at abscissa $x = 0$ in Figure 3), $m_E$ is the mobility of the charge carriers, $R_{el}$ is the electrical load, $A = H \cdot W$ is the cross-section of the toroidal coil, where $H$ and $W$ are the heigh and width of the torus section respectively, $\mu_{core}$ is the magnetic permeability of the core material, $v$ is the velocity of the neutrals, $x_1(t)$ and $x_2(t)$ are the abscissas of the two ends of the charge cloud at the instant $t$.

The total charge is:

$$Q_{tot} = \int_{x_1(t)}^{x_2(t)} \rho(\xi, t) d\xi \tag{18}$$

The charge density distribution is obtained by combining two Maxwellian distributions univocally determined, at each time $t$, by the abscissas $x_1(t)$ and $x_2(t)$ of the two ends and the abscissa $x_m(t)$ of the maximum of the charge cloud (see Figure 3).

By substituting the Equations from (13)–(17) in Equation (12) it can be re-written as follows:

$$\begin{aligned} \left[ \frac{2\pi m_E}{R} \rho(0, t) + \frac{N^2}{R_{el}} \right] \frac{d\Phi}{dt} + \frac{L}{\mu_{core} A} \Phi &= \\ = \rho(0, t) \cdot v - \frac{m_E}{2\varepsilon_{gas}} \rho(0, t) \int_0^R \int_{x_1(t)}^{x_2(t)} \frac{\gamma \cdot \xi}{[\xi^2 + \gamma^2]^{3/2}} \rho(\xi, t) \, d\gamma \, d\xi \\ - \frac{\pi R^2}{2} \frac{d}{dt} \int_0^R \int_{x_1(t)}^{x_2(t)} \frac{\gamma \cdot \xi}{[\xi^2 + \gamma^2]^{3/2}} \rho(\xi, t) \, d\gamma \, d\xi \end{aligned} \tag{19}$$

In (19), the term of second order derivative is not reported as it is negligible with respect to the others, therefore it results a first-order equation in $\Phi$, with time-variant coefficients, implicitly related to $\Phi$, which must be solved numerically.

Note that, to maximize the converted power, the voltage applied to the electrical load, which is given by the derivative of magnetic flux multiplied by the number of turns $N$, must be maximized. At the same time, a high number of turns increases the armature reaction, limiting the power conversion. A large value of the load resistance $R_{el}$ reduces the current circulating in the coil, and then also the armature reaction, but the power will be also reduced, therefore a compromise solution is mandatory. Increasing the radius R of the torus contributes to reduce the coefficient of $d\Phi/dt$ in the Equation (19) but, at the same time, the reluctance of the magnetic core increases, therefore also in this case a compromise value must be determined. The electrical mobility $m_E$ has in general a negative effect on power conversion, mainly because it causes the spreading of the charge carriers along the duct, reducing the induction effect. Starting from these considerations, in the present paper, only some parameters have been chosen to be optimized: the total charge $Q_{tot}$; the velocity of the gas $v$; the dimensions of the torus cross-section, $H$ and $W$ in Figure 3; the number of turns $N$. The other ones have been fixed.

Three conflicting objective functions have been selected: the output power $P$ delivered to the loads, to be maximized; the losses $L$ and the mass $M$ of the device, to be minimized.

The delivered power $P$ to the load resistance $R_{el}$ is calculated by

$$P = \frac{(V)_{rms}^2}{R_{el}} = \frac{N^2}{R_{el}} \cdot \left(\frac{d\Phi}{dt}\right)_{rms}^2 \tag{20}$$

where *rms* refers to the root mean square of the voltage $V$ and of the function $\frac{d\Phi}{dt}$.

The losses are due to both viscous losses in the gas and Joule losses in the wire. In a duct system the viscous losses are caused by both the viscosity of the fluid and roughness of the duct wall. These losses create a pressure drop along the duct since the pressure must work to overcome the viscous resistance. The Darcy-Weisbach Equation (21) is the most widely accepted formula for determining the energy loss in duct flows [26]. In this equation, the friction factor $f$, a dimensionless quantity, is a function of the Reynolds number and it is independent of the surface roughness of the duct if the flow is laminar. In fully turbulent flows, $f$ depends on both the Reynolds number and relative roughness of the duct wall. The friction factor is usually determined by using the Moody diagram that depicts the dimensionless Reynolds number against the friction factor and describes the corresponding fluid regimes [27]. In this work a smooth duct is assumed, hence, a friction factor $f = 0.008$ is considered.

The viscous losses $L_v$ are calculated by multiplying the specific losses by the mass flow rate $Q = \rho_{gas} \pi R^2 v$:

$$L_v = f \frac{L_t}{2R} \cdot v^2 \cdot Q = 0.008 \cdot \frac{\rho_{gas} \pi R L_t}{2} \cdot v^3 \tag{21}$$

where $\rho_{gas}$ is the density of the gas and $L_t$ is the length of duct.

The Joule losses $L_J$ depend on the passage of the electric current $i$ through the coil:

$$L_J = \frac{2N(H + W)}{\sigma_{cu} \cdot S} \cdot T i_{rms}^2 \tag{22}$$

where, $\sigma_{cu}$ is the coil conductivity, $S$ is the coils cross-section, and $i_{rms}$ is the root mean square value of the coil current. The total losses to be minimized are then:

$$L = L_v + L_J \tag{23}$$

The coil mass depends on the number of turns, so it is calculated by

$$M_{cu} = 2 \cdot N \cdot (H + W) \cdot S \cdot \rho_{cu} \tag{24}$$

where $\rho_{cu}$ is the coil density.

The iron mass depends on the dimensions of the toroidal core, then it is given by

$$M_{fe} = H \cdot W \cdot 2\pi \cdot \left(R + \frac{H}{2}\right) \cdot \rho_{fe} \tag{25}$$

where $\rho_{fe}$ is the core density.

The total mass to be minimized is then:

$$M = M_{cu} + M_{fe} \tag{26}$$

Note that, the previous calculations are referred to a single magnetic core, hence, the three objective functions must be doubled to obtain the values of the entire device.

## 4. Implementation and Results

In Figure 4, the schematics of the Simulink model that solve Equation (19) is reported. The subscripts 0, 1, 2, and $m$ in the schema indicate the abscissas 0, $x_1(t)$, $x_2(t)$ and $x_m(t)$ in Figure 3. To determine the density distribution function $\rho(x,t)$, these three abscissas must be evaluated. To this end, two electrical fields are calculated in correspondence of the three abscissas: the fields due to the charge distribution ($E_{\rho1}$, $E_{\rho m}$, $E_{\rho2}$) and those due to the time variant flux ($E_{\phi1}$, $E_{\phi m}$, $E_{\phi2}$). The sum of these two fields gives the electrical field $E_{TOT}$ contribution to the velocity of charges, which is combined with the drag velocity of neutrals, providing the charge velocities ($v_1$, $v_m$, $v_2$) in the three abscissas. By integrating these velocities, starting from the initial values $x_1(0)$, $x_m(0)$ and $x_2(0)$, the three abscissas are obtained, and the charge distribution is defined. The initial charge distribution $\rho(x,0)$ is assumed as Gaussian (the charge cloud not yet deformed by the field) with $x_m(0) = mean(x_1(0), x_2(0))$.

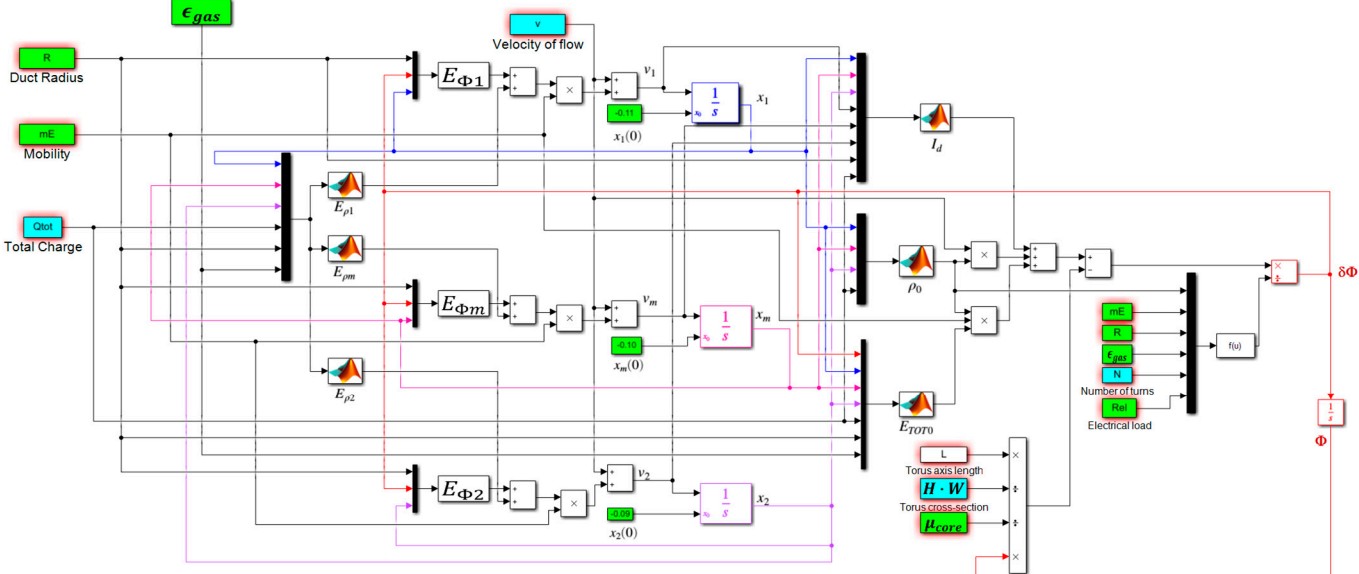

**Figure 4.** Scheme of the model developed in Simulink environment (green boxes: fixed design parameters; cyan boxes: design parameters of the MO-TS problem).

Once velocity and distribution of the charges are known, it is possible to calculate the three components of current in the origin $x = 0$, namely the displacement current ($I_d$), the current ($\rho_0 \times v$) due to the dragging of the charges in the origin, and the current due to the electrical field in the origin ($E_{TOT0}$). Then, the derivative of the flux $\delta\Phi = d\Phi/dt$ is evaluated, from which the voltage and the power evolutions (see Equation (20)), and finally the flux $\Phi$ can be determined.

To perform a single Simulink integration of Equation (19) a time interval of 5 ms is set. This interval has been chosen to avoid each discharge interferes with the following one. Consequently, the rate $f_{dis} = 200$ discharges per second is assumed.

In Figure 4, the chosen design parameters in input to the optimization procedure are highlighted in cyan color and reported in Table 1 together with their feasible ranges, identified from the preliminary studies reported in [11]. In the same Figure 4, the fixed parameters are marked as green boxes and reported in Table 2 together with the assigned values still referring to [11].

**Table 1.** Design parameters of the MO-TS problem.

| Design Parameter | Quantity | Range | Units |
|:---:|:---:|:---:|:---:|
| $Q_{tot}$ | Total charge | [0.1–1.1] | $10^{-3}$ [C] |
| $v$ | Velocity of flow | [150–400] | [m·s$^{-1}$] |
| $H$ | Height torus section | [0.1–0.5] | [m] |
| $W$ | Width torus section | [0.1–0.5] | [m] |
| $N$ | Number of turns | [3–11] | [a.u.] |

**Table 2.** Other fixed design parameters.

| Design Parameter | Quantity | Value | Units |
|:---:|:---:|:---:|:---:|
| $S$ | Coils cross-section | $2.5 \times 10^{-6}$ | [m$^2$] |
| $R$ | Radius of the duct | 0.1 | [m] |
| $L_t$ | Length of the duct | 0.01 | [m] |
| $\mu$ | Air viscosity (1226 °C) | $1.81 \times 10^{-5}$ | [Pa·s] |
| $\rho_{gas}$ | Gas density | 7.67 | [kg·m$^{-3}$] |
| $\mu_{core}$ | Core permeability | $6.28 \times 10^{-2}$ | [H·m$^{-1}$] |
| $\varepsilon_{gas}$ | Gas permittivity | $8.85 \times 10^{-12}$ | [F·m$^{-1}$] |
| $m_E$ | Electrical mobility | $2 \times 10^{-4}$ | [m$^2$·V$^{-1}$·m$^{-1}$] |
| $R_{el}$ | Electrical load | 150,000 | [$\Omega$] |
| $\sigma_{cu}$ | Coil conductivity | $5.95 \times 10^7$ | [S·m$^{-1}$] |
| $\rho_{cu}$ | Coil density | 8960 | [kg·m$^{-3}$] |
| $\rho_{fe}$ | Core density | 7860 | [kg·m$^{-3}$] |

Summarizing, the multi-objective optimization problem can be formalized as in the following:

$$
\begin{cases}
\displaystyle\min_{x\in\mathbb{R}^3} fobj = -(fit_1(\boldsymbol{F}) + fit_2(\boldsymbol{F})) \\
\quad\quad Subject\ to \\
\quad\quad \boldsymbol{F} \in \wp F(\boldsymbol{x}_c)
\end{cases}
\tag{27}
$$

where

$$
\wp\mathcal{F}(\boldsymbol{x}_c) \succ
\begin{cases}
\boldsymbol{F} = \left[\displaystyle\min_{x\in\mathbb{R}^5}L(\boldsymbol{x}), \min_{x\in\mathbb{R}^5}M(\boldsymbol{x}), \max_{x\in\mathbb{R}^5}P(\boldsymbol{x})\right]^T \\
\boldsymbol{x} = [Q_{\text{tot}}, v, H, W, N]^T \\
\boldsymbol{x} \in neighbour(\boldsymbol{x}_c) \\
\quad Subject\ to \\
\begin{bmatrix} 0.1 \\ 150 \\ 0.1 \\ 0.1 \\ 3 \end{bmatrix} \leq \boldsymbol{x} \leq \begin{bmatrix} 1.1 \\ 400 \\ 0.5 \\ 0.5 \\ 11 \end{bmatrix} \begin{bmatrix} C \\ m \cdot s^{-1} \\ m \\ m \\ a.u. \end{bmatrix}
\end{cases}
\tag{28}
$$

In the optimization algorithm, at each iteration, two fitness values are assigned to all the solutions $\boldsymbol{x}_i$ belonging to the neighbor of the current solution $\boldsymbol{x}_c$. The first one allows to store in the $\wp\mathcal{F}$ a new $\boldsymbol{F}(\boldsymbol{x}_{nd})$ that is non dominated ($\succ$) by those present until then, and remove these latter ones:

$$
fit_1 = \frac{1 + N_d(\boldsymbol{x}_i)}{1 + N_{nd}(\boldsymbol{x}_i)}
\tag{29}
$$

where $N_d(\boldsymbol{x}_i)$ and $N_{nd}(\boldsymbol{x}_i)$ are the number of elements in $\wp\mathcal{F}(\boldsymbol{x}_c)$ that are dominated or not by $\boldsymbol{x}_i$, respectively.

The second fitness function favors the chosen non dominated points to be distributed in a regular grid on $\wp\mathcal{F}$:

$$fit_2 = e^{-0.5\left(\frac{d-mean}{std\_dev}\right)^2} - e^{-0.5\left(\frac{d}{std\_dev}\right)^2} \tag{30}$$

where $d$ is the smallest distance between $\boldsymbol{F}(\boldsymbol{x}_i)$ and the points in $\wp\mathcal{F}$, and *mean* and *std_dev* are parameters determining the sampling step of the Pareto Front. Finally, the objective function to be minimized is:

$$f_{obj} = -(fit_1 + fit_2) \tag{31}$$

Algorithm 1 reports the pseudo-code of the MO-TS utilized in the present paper.

---

**Algorithm 1:** MO-TS

---

    **Input**:
        Domain $S$ of feasible solutions $\boldsymbol{x} \in \mathbb{R}^n$
    **Output**:
1:      Regularly sampled Pareto front $\wp\mathcal{F}$
2:  % Initialization
3:  Randomly select the starting solution $\boldsymbol{x}_c \in S$
4:  $\boldsymbol{F}(\boldsymbol{x}_c) \leftarrow \begin{bmatrix} \boldsymbol{L}(\boldsymbol{x}_c) & \boldsymbol{M}(\boldsymbol{x}_c) & \boldsymbol{P}(\boldsymbol{x}_c) \end{bmatrix}$ % $L$ = Losses; $M$ = Mass; $P$ = Power
5:  $\wp\mathcal{F} \leftarrow \boldsymbol{F}(\boldsymbol{x}_c)$      % Initialize the Pareto front $\wp\mathcal{F}$ with the point $\boldsymbol{F}(\boldsymbol{x}_c)$
6:  **while** #It < $N_{MO-TS}$ and #pnts < $N_{PF}$:
7:     % New cycle of Tabu Search
8:     $TL \leftarrow \boldsymbol{x}_c$ % Initialize the Tabu List TL with the starting solution $\boldsymbol{x}_c$
9:     **while** #it < $N_{TS}$ **and** #objcalls < $N_{obj-calc}$:
10:        % Generate a neighbourhood $\{\boldsymbol{x}_i\}$ of $\boldsymbol{x}_c$
11:        **for** $N_n$ **times**: % $N_n$ = max elements of the neighbour
12:          Select a candidate $\hat{\boldsymbol{x}} \notin TL$ near $\boldsymbol{x}_c$
13:          Move $\hat{\boldsymbol{x}}$ to the Tabu List $TL$ for next $TT$ moves
14:          % $L$ = Losses; $M$ = Mass; $P$ = Power
15:                $\boldsymbol{F}(\hat{\boldsymbol{x}}) \leftarrow \begin{bmatrix} \boldsymbol{L}(\hat{\boldsymbol{x}}) & \boldsymbol{M}(\hat{\boldsymbol{x}}) & \boldsymbol{P}(\hat{\boldsymbol{x}}) \end{bmatrix}$
16:          % fitness of the point $\hat{\boldsymbol{x}}$
17:          $\boldsymbol{f}(\boldsymbol{F}(\hat{\boldsymbol{x}})) \leftarrow -(\boldsymbol{fit_1}(\boldsymbol{F}(\hat{\boldsymbol{x}})) + \boldsymbol{fit_2}(\boldsymbol{F}(\hat{\boldsymbol{x}})))$
18:          Add $\hat{\boldsymbol{x}}$ to $\{\boldsymbol{x}_i\}$ according to ***Reactive TS***
19:        **end for**
20:        $\boldsymbol{x}_c \leftarrow \boldsymbol{argbest}(\boldsymbol{f}(\boldsymbol{x}_i))$ % Select the $\boldsymbol{x}_i$ with the best fitness
21:     **end while**
22:     **if** $\boldsymbol{F}(\boldsymbol{x}_j) \succ \boldsymbol{F}(\boldsymbol{x}_c) \,\forall \boldsymbol{x}_j \,\big|\, \boldsymbol{F}(\boldsymbol{x}_j) \in \wp\mathcal{F}$ % if any Pareto point dominates $\boldsymbol{x}_c$
23:        Discard $\boldsymbol{x}_c$
24:     **else**
25:        Add $\boldsymbol{F}(\boldsymbol{x}_c)$ to $\wp\mathcal{F}$
26:        % If there are Pareto points dominated by $\boldsymbol{x}_c$
27:        **if** $\boldsymbol{F}(\boldsymbol{x}_c) \succ \boldsymbol{F}\big(\boldsymbol{x}_j\big) \,\forall \boldsymbol{x}_j \,\big|\, \boldsymbol{F}\big(\boldsymbol{x}_j\big) \in \wp\mathcal{F}$
28:          Discard $\boldsymbol{x}_j$
29:        **end if**
30:     **end if**
31: **end while**
    **return** $\wp\mathcal{F}$

---

In the MO-TS algorithm a solution $\boldsymbol{x}$ is randomly selected among the feasible solutions $S$ and it is assumed as current solution $\boldsymbol{x}_c$. A Tabu List TL and a Pareto Front list $\wp\mathcal{F}$ are firstly initialized as empty lists. The most promising neighborhood of the $\boldsymbol{x}_c$ is deeply explored using a Reactive TS scheme [25] and the vector of the objective functions $\boldsymbol{F}(\boldsymbol{x})$ is evaluated for each neighbor solution, not present in the TL, using the combination of Simulink model and MATLAB functions. The objective function $f_{obj}$ is then evaluated by referring to the current $\wp\mathcal{F}(\boldsymbol{x}_c)$. The best solution is then assumed as new current solution

$x_c$ and the TL is updated. If $F(x_c)$ satisfies the fitness functions, also the Pareto Front is updated, adding the current point, and removing all the points dominated by the current one. These steps are iterated until a stop criterion is met. The current $\wp\mathcal{F}$ is then assumed as the final one, which contains points distributed in a regular grid.

Table 3 reports the MO-TS algorithm's parameters.

**Table 3.** MO-TS parameters.

| Parameter | Description | Value |
|---|---|---|
| $N_{PF}$ | Size of Pareto Front | 150 |
| *mean* | Mean | 500 |
| *std_dev* | Standard Deviation | 167 |
| $N_{MOTS}$ | Max number of MO-TS iterations | 200 |
| $N_{TS}$ | Max number of TS iterations | 750 |
| $N_{obj-calc}$ | Max number of objective function calculations | $1 \times 10^4$ |
| $N_n$ | Max elements of the neighbor | 129 |
| $TT$ | Range of Tabu Tenure | 4–30 |

Note that, the MO-TS algorithm is easy to be customized to different optimization problems, when the design, materials, and/or the operating modes are significantly different from the ones under consideration, providing a new SIMULINK model, or, if necessary, a FEM model, is implemented.

All the calculations in this paper have been performed using a XEON, 60 core, 2.5 GHz, with 110 GB of RAM running under Windows 10 Professional. Some algorithm's parameters have been empirically chosen. As reported in Table 3, in the MO-TS algorithm, the list of the Pareto front $\wp\mathcal{F}$ has been sized to store 150 non-dominated solutions and the *mean* and the *std_dev* parameters in (30) are set equal to 500 and 167 respectively. The algorithm stops when the $\wp\mathcal{F}$ list is full or when the maximum number of 200 MO-TS iterations is reached. The stop criterion of each TS cycle is the maximum number of TS cycles, set equal to 750, or the maximum number of objective function calculations, set equal to 10,000.

The optimization procedure returns a sampling of the Pareto front (Figure 5) composed by 93 Pareto optimal solutions after 200 iterations of MO-TS performing a total number of $2 \times 10^6$ objective function evaluations, each lasting 500 ms. Even if the optimization procedure may seem to require too long a computation time, this is however negligible if compared to the time of a common design optimization of such complex devices. Furthermore, the proposed MO-TS could be easily parallelized with a considerable reduction in time. Such a parallel implementation could allow the extension of the optimization to further design parameters.

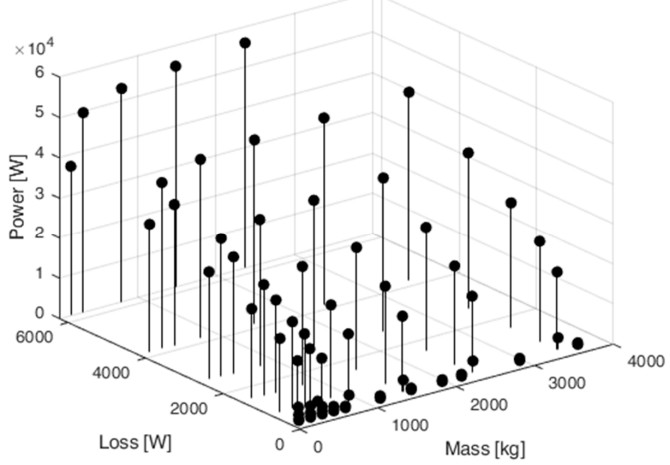

**Figure 5.** 3-D Pareto Front.

As can be noted, the algorithm is able to find a uniformly sampled Pareto optimal solutions that spam in a range of [148–3704] kg of the mass, [325–6234] W of losses, and [0.5–55,911] *W* of output power.

Further equality and inequality constraints can be used to choose among the optimal solutions, such as minimum allowed weight (depending on the application field) and/or the maximal achieved performance, and so on.

In Table 4, some Pareto solutions selected from the $\wp\mathcal{F}$ and the corresponding design parameters values are reported. The $\wp\mathcal{F}$ represents a general framework of possible design solutions. The final choice must be guided by the specific application context in which the machine is designed.

**Table 4.** Values of the three objective functions and of the corresponding design parameters.

| | Power $P$ [kW] | Loss $P$ % | Mass [kg] | $Q_{tot}$ [C] | $v$ [m·s$^{-1}$] | $H$ [m] | $W$ [m] | $N$ |
|---|---|---|---|---|---|---|---|---|
| 1 | 55.9 | 11.15 | 2370 | $1.1 \times 10^{-3}$ | 400 | 0.4 | 0.2 | 3 |
| 2 | 47.0 | 8.93 | 3457 | $1.1 \times 10^{-3}$ | 350 | 0.5 | 0.2 | 3 |
| 3 | 38.7 | 6.87 | 3457 | $1.1 \times 10^{-3}$ | 300 | 0.5 | 0.2 | 3 |
| 4 | 31.2 | 4.96 | 3457 | $1.1 \times 10^{-3}$ | 250 | 0.5 | 0.2 | 3 |
| 5 | 29.4 | 5.24 | 790 | $1.1 \times 10^{-3}$ | 250 | 0.2 | 0.2 | 3 |
| 6 | 26.5 | 10.00 | 148 | $1.1 \times 10^{-3}$ | 300 | 0.1 | 0.1 | 3 |
| 7 | 22.2 | 6.97 | 148 | $1.1 \times 10^{-3}$ | 250 | 0.1 | 0.1 | 3 |
| 8 | 17.9 | 1.94 | 790 | $1.1 \times 10^{-3}$ | 150 | 0.2 | 0.2 | 3 |
| 9 | 14.6 | 2.44 | 148 | $1.1 \times 10^{-3}$ | 150 | 0.1 | 0.1 | 3 |
| 10 | 3.2 | 10.5 | 148 | $0.5 \times 10^{-3}$ | 150 | 0.1 | 0.1 | 3 |

The first relevant result is that all the solutions of $\wp\mathcal{F}$ have a number of turns equal to the minimum feasible. This result is not surprising, since if it is true that the number of turns increases the value of the voltage applied to the load, and therefore the power delivered, but it is also true that the armature reaction increases, effectively reducing the primary current flowing in the tube. At the same time, the increase in the number of turns increases the losses due to the Joule effect, as a constant wire section has been assumed.

An examination of the front solutions shows that power values > 13 kW always require the maximum value of the electric charge $Q_{tot}$ in the admissibility range. The electric charge has not been included among the objectives to be optimized, but this does not mean that all the admissible values are equivalent. The designer, in fact, will tend to choose solutions with the lowest possible values of the charge, because the generation and containment of the charge require technical solutions that increase the complexity and cost of the device. It will therefore be up to the designer to take this constructive aspect into account when choosing the solution to implement. For powers $P$ > 18 kW, the electric charge being fixed, the $\wp\mathcal{F}$ provides solutions either with large core dimensions, and therefore with large mass, or with high gas velocities, and therefore with high viscous losses.

The first solution reported in Table 4 is the one with the highest power among those contained in $\wp\mathcal{F}$. Against a power output of 55.9 kW, the losses amount to about 11% of the power output (net useful power 49.7 kW), due to the high speed of the gas (400 m/s). In the same table, the second solution shows a lower power (47 kW) in which the losses are equal to 8.9% of the power supplied (net useful power 42.8 kW), thanks to the fact that it uses a gas speed lower than the previous one (350 m/s). At the same time, the mass of the device is significantly greater than that used in the first solution.

By accepting even lower power values (38.7 kW), with the same dimensions and therefore mass, by reducing the speed to 300 m/s, the percentage losses are reduced to 6.87% (case no. 3, net useful power 36.0 kW). By further reducing the speed to 250 m/s and keeping the dimensions unchanged, the power output is reduced to 31.2 kW with losses equal to 4.96% (net useful power 29.6 kW).

Continuing to analyze the solutions in Table 4, it can be observed that, with a modest reduction in the delivered power (29.4 kW) and a slight increase in the percentage losses (5.24%, net useful power 27.8 kW), it is possible to reduce the mass to less than $1/4$ of that of the previous solution.

The solution no. 6 is particularly interesting for mobile applications. With a power output of 26.5 kW and losses of 10% (net useful power 23.9 kW) it has a mass of 148.2 kg. The solution no. 7 with the same geometry and with a gas velocity of 250 m/s delivers a power of 22.2 kW with losses that are reduced to 7% (net useful power 20.7 kW).

The solution no. 8 is the one with the lowest percentage value of the lost power. Against 17.9 kW of power output, the losses amount to 0.347 kW (1.94%), with a net useful power of 17.6 kW. In comparison, the solution no. 9 delivers a power of 14.6 kW and 2.44% of losses (net useful power 14.3 kW), but it has a significantly lower mass than the previous solution, so it is potentially more suitable for mobile applications.

Changing the power scale, solution no. 10 delivers 3.2 kW with 10.5% of losses (net useful power 2.9 kW). The mass is of 148 kg, therefore it is suitable for small mobile applications. Furthermore, it can be seen that the required charge $Q_{tot}$ is much lower than that one required for the previously examined solutions.

In solutions with power lower than 3 kW, the losses assume percentage values which render the proposed technology unfeasible.

## 5. Conclusions and Future Work

In this paper, a procedure is presented for the optimization of the design of an inductive magneto-hydro-dynamic electric generator. The layout of the device has kept the advantages of the conventional MHD generators, the main one being the static conversion of the energy, but at the same time it avoids the shortcomings of this kind of process, typically the small range of operative temperatures, the need of superconducting coils, the corrosion of the electrodes. The novelty of the technology means that there are no standards for the design. In the present paper, the optimal design of the MHD generator has been performed by implementing a multi-objective Tabu Search algorithm to address the multiple conflicting objectives of the problem. The algorithm returns the Pareto optimal solutions that allows to analyze different scenarios fitting further requirements of the specific application. For example, in space applications or for automotive the strictest requirements are mass and size, whereas for industrial plants, weight could be less important to benefit delivered power.

Due to the exponentially increase of the algorithm complexity with the dimension of the search space, the optimization has been limited to some design parameters, whereas the others have been empirically fixed. As discussed in the previous sections, different design parameters can produce the same effect on some objectives, so a different choice can be done. For example, the previous analysis shows that the number of coils in the different Pareto optimal solutions assumes the minimum value in its range and it could be removed from the set of input design variables and substituted with another one, such as the length of the duct. Note that, the implemented MO-TS lends itself well to studying different input spaces without modification of its schema.

Future works will focus on relaxing some assumptions made in this paper. Firstly, the choice of a one-dimensional SIMULINK model of the MHD electric generator, used to evaluate the objective functions, which implies a number of approximations. The most important concerns the fact that the velocity profile throughout the cross-section of the duct is neglected. In fact, the charge carriers due to mutual repulsion tend to distribute themselves in the region close to the walls, where the velocity is lower, ultimately reducing the intensity of the ionic current. In future, an effective velocity of the charges lower than that of the neutrals will be considered. Moreover, it has been assumed that the magnetic induction **B** is constant along the entire cross section of the magnetic core, i.e., it has been assumed to neglect its decay as the distance from the duct axis increases. The effect of this assumption becomes important especially if the device works near iron

saturation conditions, and will be relaxed in future. Another noteworthy aspect concerns the impedance of the load, which for simplicity has been assumed purely resistive. To improve the converted power, it will be necessary in future to adapt the load to the generator.

**Author Contributions:** Conceptualization, A.M. and S.C.; methodology, A.F.; software, S.C. and A.M.; validation, S.C., A.M. and A.F.; formal analysis, S.C., A.M. and A.F.; investigation, S.C., A.M. and A.F.; writing—original draft preparation, A.M.; writing—review and editing, S.C., A.F. and A.M. All authors have read and agreed to the published version of the manuscript.

**Funding:** This research was developed within the project "Si.E.S.-Sistema Energetico Sostenibile", funded by Autonomous Region of Sardinia (Italy), under the regional law No 19/96.

**Data Availability Statement:** Not applicable.

**Conflicts of Interest:** The authors declare no conflict of interest.

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
