# Peer review of "Optimal Design of an Inductive MHD Electric Generator"

_sustainability, doi:10.3390/su142416457_

Round 1

Reviewer 1 Report

Below are the comments and suggestions which, when appropriately addressed by the authors, may enhance the quality of the paper:

1.      Please specify the unique contributions of this paper.

2.      The article would be appropriate to explicitly indicate the scientific benefits of the present article explicitly.

3.      A pseudocode is a nice-to-have the illustrate the algorithm

4.      Authors should discuss in more detail the selected references that are currently cited collectively.

5.      Authors should give the details of algorithm parameters setting

Author Response

The authors wish to thanks the Reviewers for their very useful comments and their suggestions that allow us to improve the paper. In the attached file the responses to the comments are reported. In black the Reviewers’ comments and in red our answers. The revised version of the paper contains the new text in red.

Reviewer 2 Report

Please read the attached file. Thank you. 

Author Response

(The authors gave the same response as above.)

Reviewer 3 Report

(overall assessment) As regards plasma reactors, a simplified approach with ‘lumped parameters’ is set up for simulations on SIMULINK (with its 1-D model) to reach optimized design and operational conditions. This manuscript is worthy of a publication when the following issues are properly addressed.

(question 1) What is the major distinctions of this manuscript from the authors’ recent publications: [1] . Carcangiu, S.; Forcinetti, R.; Montisci, A. Simulink Model of an Inductive MHD Generator. Magnetohydrodynamics 2017, 53, 255– 509 265, doi:10.22364/mhd.53.2.4.  and [2] Carcangiu, S.; Fanni, A.; Montisci, A. Multiobjective Tabu Search Algorithms for Optimal Design of Electromagnetic Devices. 515 IEEE Trans. Magn. 2008, 44, 970–973, doi:10.1109/TMAG.2007.916336. 

(comment 1) The analytical formulas presented in the manuscript are rather simplified from the original set of the time-dependent governing equations where both electric and magnetic fields are coupled among them (namely, the Maxwell equations) in interaction with other phenomena such as fluid flows and ion dynamics etc.

(request 1) Please make somewhere a summary of underlying assumptions in reaching the simplified formulas.

(question 2) How to deal with the optimization problem when the design, materials, and/or the operating modes are significantly different from the ones under consideration?

Author Response

(The authors gave the same response as above.)

Round 2

Reviewer 1 Report

The authors discussed the Multi-objective Optimization Algorithm in section 3.1, thus I suggested the authors discuss more literatures in this area, For instance, you can refer to genetic algorithm (https://doi.org/10.1016/j.jclepro.2019.118845), hybrid algorithm (https://doi.org/10.1016/j.asoc.2020.106739 and https://doi.org/10.1016/j.cie.2020.106768)

Reviewer 3 Report

The authors made several modifications and additions to address my concerns. They made it clear that their work is a significant improvement over their recent publishes papers.

They also added basic assumptions underlying their work.